# Multicolor ultralong phosphorescence from perovskite-like octahedral α-AlF₃

Peisheng Cao[1], Haoyue Zheng[2] & Peng Wu ◉[1,2]✉

Designing organic fluorescent and phosphorescent materials based on various core fluorophore has gained great attention, but it is unclear whether similar luminescent units exist for inorganic materials. Inspired by the $BX_6$ octahedral structure of luminescent metal halide perovskites (MHP), here we propose that the $BX_6$ octahedron may be a core structure for luminescent inorganic materials. In this regard, excitation-dependent color-tunable phosphorescence is discovered from α-AlF₃ featuring AlF₆ octahedron. Through further exploration of the $BX_6$ unit by altering the dimension and changing the center metal (B) and ligand (X), luminescence from KAlF₄, (NH₄)₃AlF₆, AlCl₃, Al(OH)₃, Ga₂O₃, InCl₃, and CdCl₂ are also discovered. The phosphorescence of α-AlF₃ can be ascribed to clusterization-triggered emission, i.e., weak through space interaction of the $n$ electrons of F atoms bring close proximity in the AlF₆ octahedra (inter/intra). These discoveries will deepen the understanding and contribute to further development of $BX_6$ octahedron-based luminescent materials.

Luminescent materials are indispensable for our daily life, especially in lighting, displaying, and imaging-related applications[1,2]. Therefore, luminescent materials design (either organic or inorganic) is of great importance and attracts great attention. It is widely accepted that luminescence from organic materials can be ascribed to their core structure in most cases (Fig. 1, together with the substituents), for example, fluorescent materials from xanthene[3–7] and phosphorescent materials based on carbazole[8–11]. Such core structure endow organic fluorophores with great flexibility and processability. While for inorganic luminescent materials, although structurally diverse and mostly acting as host materials for doping of transition- or rare-earth metal ions (e.g., ZnS[12] and SrAl₂O₄[13]), core structure as the light-emitting unit has seldom been reported and explored like organics. So, is there similar core structure for the luminescent inorganic materials?

All-inorganic metal halide perovskites (MHPs), a type of semiconductor materials with excellent photoelectric properties[14], have been widely used in solar cells[15], LED[16], and thermoelectric modules[17]. The core structure of luminescent MHPs can be described as the $BX_6$ octahedron (Fig. 1), which is constituted by the central cation (B, hexa-coordinated) and six halide ligands (X=Cl, Br, I). Normally, the $BX_6$ octahedron is organized in an all-corner-sharing 3D network. Due to the adjustable octahedral connectivity, a series of lower dimensional

metal halide-based luminescent perovskite derivatives have been reported[18]. On the other hand, the central cation and ligand halides could be altered, leading to tunable luminescence performance from 3D and lower dimension metal halides[19,20]. Therefore, the $BX_6$ octahedron is an important structure for the luminescence of MHPs, but whether such unit can be generalized for other luminescent inorganic materials remains unexplored.

In this work, we find that inorganic materials constructed by the $BX_6$ octahedron exhibit interesting long-lived room temperature phosphorescence (RTP), i.e., the $BX_6$ octahedron may be regarded as a basic unit for the luminescent inorganic materials. For example, α-AlF₃, the material constructed by AlF₆ (the lightest $BX_6$ octahedron) with 3D perovskite-like structure[21], shows color-tunable RTP (up to 7 s for the blue emission). When lowering the dimension of the AlF₆ octahedron, luminescence from KAlF₄ (2D) and (NH₄)₃AlF₆ (0D) is also discovered. Moreover, by changing B and X in the octahedral emissive unit of $BX_6$, luminescence from AlCl₃, Al(OH)₃, Ga₂O₃, InCl₃, and CdCl₂ are also obtained. Similar to $BX_6$ octahedron-based MHPs, the luminescence from AlF₃ exhibit typical self-trapped exciton (STE) emission. Besides, the octahedron also brings close proximity of F atoms, resulting in weak through-space interaction of the $n$ electrons in F atoms for clusterization-triggered emission (CTE, recently found in $n$ electron-rich

[1]College of Chemistry, Sichuan University, Chengdu 610064, China. [2]Analytical & Testing Center, Sichuan University, Chengdu 610064, China.
✉e-mail: wupeng@scu.edu.cn

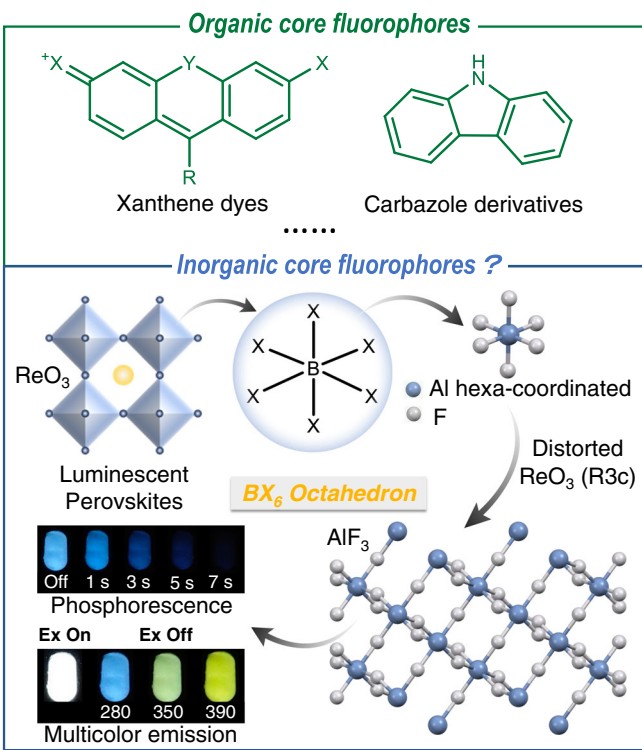

**Fig. 1 | Schematic illustration of luminescent materials design.** Inspired by the general luminescent material design based on organic core fluorophores and the widely investigated metal halide perovskites, here, we proposed $BX_6$ octahedral unit as inorganic luminescent core structure.

organics for excitation-dependent color-tunable phosphorescence[22–24]. It should be noted that it is normally the heavy atoms that drives the formation of triplet and phosphorescence in inorganics (e.g., previous Al-based luminescent materials, Supplementary Table 1). However, α-$AlF_3$ contains only light elements. Therefore, the discovery here is interesting for both organic and inorganic phosphors. In addition, the intriguing color-tunable phosphorescence without extra sophisticated molecular design can be explored for facile UV light detection with visible colored afterglow emission as readout.

## Results

### Luminescence of α-$AlF_3$

To investigate the luminescent properties of the $BX_6$ octahedron, hexa-coordinated $AlF_6$ was chosen first. Considering its outermost electronic structure ($3s^2 3p^1$), aluminum is lightest atom to generate the octahedral structure. Meanwhile, $F^-$ is the smallest anion[25] that can coordinate with $Al^{3+}$. The corner shared octahedra of $AlF_6$ results in the formation of three-dimensional network of α-$AlF_3$ (Fig. 1). Normally, $AlF_3$ is used as the electrolyte regulator in the aluminum smelting industry for increasing the melting point and conductivity. However, its photophysical properties are rarely studied.

Here, we found $AlF_3$ exhibited exciting color-tunable luminescence at room temperature (Fig. 2a and Supplementary Fig. 3), irrespective of its origins (Supplementary Table 6 and Fig. 2). From the time-resolved emission spectra (TRES, Fig. 2b and Supplementary Fig. 11), the luminescence of α-$AlF_3$ could be attributed to phosphorescence, with quantum yield ($\Phi_P$) of ~4.22% and lifetime up to ~0.9 s ($\lambda_{ex}$ = 280 nm, Fig. 2c). The room-temperature afterglow of α-$AlF_3$ could last more than 7 s (naked eye observable), and the excitation-dependent blue to yellow afterglow could be visualized clearly after ceasing the excitation (Fig. 2d, Supplementary Figs. 12–13, and Supplementary Movies 1–3). The color-tunable emission was also clearly

revealed by the Commission Internationale de l'Eclairage (CIE) chromaticity coordinates (Fig. 2e). In addition, pure white light emission could be obtained (approaching CIE of 0.33, 0.33) when changing $\lambda_{ex}$ from 270 nm to 390 nm (Fig. 2d and Supplementary Fig. 15).

To exclude the potential influence from trace impurities, direct synthesis of $AlF_3$ through exposing aluminum metal of the highest purity available to HF vapor was carried out. As expected, similar emission properties were also obtained (Supplementary Figs. 6-7), confirming that the luminescence was exactly from $AlF_3$. Furthermore, the purchased and as-prepared samples were processed by calcination, ball milling, and acid-washing, and no appreciable change of the luminescence property was received (Supplementary Figs. 8-10).

### Extending of the $BX_6$ octahedral luminescent unit

Since the 3D perovskite-like structure of α-$AlF_3$ is constituted by the $AlF_6$ octahedron, the basic unit of $BX_6$ was further explored by adjusting the dimension of the octahedra (Fig. 3a). As shown in Figs. 3b and 3c, phosphorescence from $KAlF_4$ (2D, layers of corner-sharing $AlF_6$ octahedra and BCC coordinated $K^+$) and $(NH_4)_3AlF_6$ (0D, isolated $AlF_6$ octahedra surrounded by $NH_3$ ligands) was also collected (Supplementary Figs. 22 and 25), accompanied with similar excitation-dependent emission. The crystalline structure information (Supplementary Table 7) and XRD patterns confirmed that both $KAlF_4$ and $(NH_4)_3AlF_6$ were composed by the $AlF_6$ octahedral unit and belonged to 2D and 0D metal halides structures (Supplementary Fig. 20), respectively. It should be noted that lowering the dimension of the octahedra resulted in largely decreased phosphorescence intensity and shortened lifetime (Fig. 3c), indicating that the linkage of the $AlF_6$ octahedra also contributed to the observed phosphorescence.

Next, the ligand (X) of the $BX_6$ basic unit was changed with other halogens, namely $AlCl_3$ and $AlBr_3$ (Fig. 3a). As shown in Fig. 3d and Supplementary Fig. 23, $AlCl_3$ (cubic layered $AlCl_6$ octahedra) still exhibited appreciable long-lived phosphorescence (Fig. 3e and Supplementary Fig. 26 and Supplementary Movie 4). Through altering halide composition from F to Br, their emission spectra are readily tunable from blue to yellow (Insets of Fig. 3d), which was similar with the emission tunable perovskite materials through halide engineering[26]. Although the outer space of Al is unable to accommodate six Br atoms to form stable hexadentate structure due to their relatively large atom radius discrepancy, $AlBr_3$ (tetrahedral) is still luminescent, but with significantly reduced intensity and shortened lifetime (Fig. 3d, 3e). When altering the ligand from F to O, phosphorescence from the $AlO_6$ (octahedral) derivatives ($Al(OH)_3$ or $Al_2O_3$, 2D sheets of edge sharing $Al(OH)_6$ octahedra) was also collected (Fig. 3d and Supplementary Fig. 23).

Upon changing the metal center of $BX_6$, octahedral structure from $Ga_2O_3$ (mixture of $GaO_6$ octahedral and $GaO_4$ tetrahedral), $InCl_3$ (distorted $InCl_6$ octahedra in the 1D chains), and $CdCl_2$ (2D sheets of edge-sharing $CdCl_6$ octahedra) can be expected. Again, phosphorescence from these species was successfully collected (Fig. 3f and Fig. 3g, Supplementary Figs. 23 and 27, and Supplementary Movie 5 for $Ga_2O_3$ afterglow). It should be noted that there are diverse hexa-coordinate structures evolved from different bond angles (X-B-X), thus varied phosphorescence properties (excitation, emission, intensity, and lifetime, Fig. 3h)[27].

### Luminescence mechanism of α-$AlF_3$

The luminescence mechanism of the octahedral unit was investigated with the 3D network $AlF_6$ (α-$AlF_3$). Compared with MHPs, α-$AlF_3$ showed the 3D perovskite-like structure, but without the insertion of alkali metal cations. These cations only contribute to lattice stabilization and does not participate in the formation of the frontier molecule orbitals[28]. The intrinsic emission of MHPs could be originated from free, bound and self-trapped excitons[29]. Among them, self-trapped exciton (STE) emission is a well-accepted mechanism to account for

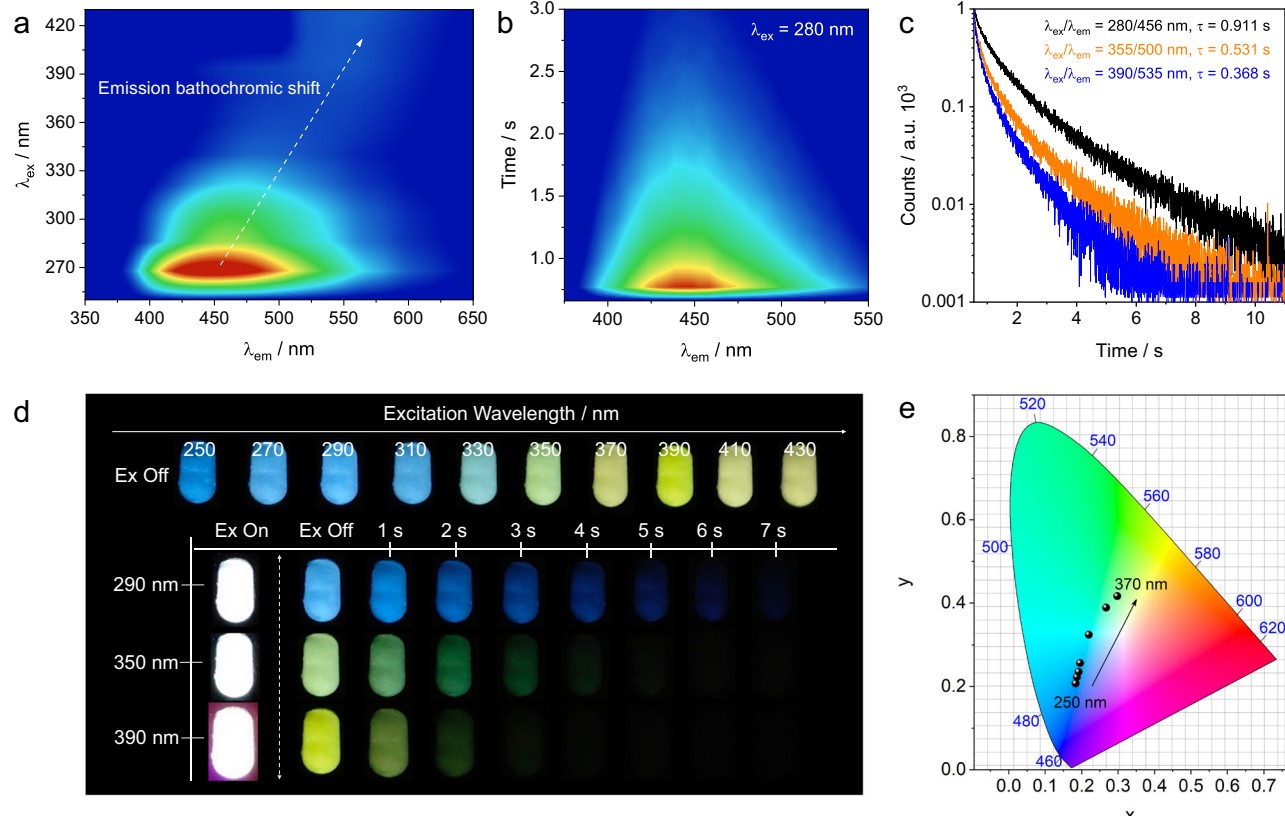

**Fig. 2 | Luminescence properties of α-AlF₃ (calcinated from AlF₃·3H₂O).**
**a** excitation- phosphorescence emission mapping of AlF₃ (delay time: 40 ms);
**b** time-resolved emission spectra (TRES, $\lambda_{ex}$ = 280 nm) of AlF₃; **c** lifetime decay
profiles of phosphorescent emission excited at 280, 355, and 390 nm, respectively;
**d** photographs taken under different excitation (250 to 430 nm) off and afterglow
emission images excited at 290, 350 and 390 nm, respectively (the excitation-
dependent afterglow can be observed in the excitation range from 250 to 510 nm at
room temperature condition); **e** CIE coordinates of AlF₃ phosphorescence under
different excitation (250 to 370 nm).

the broadband and large-stokes shift emission[30–33]. For α-AlF₃, similar
broad-band excitation-dependent blue to yellow emission was
emerged, with full width at half-maximum (FWHM) of 120 nm
and Stokes shift up to -180 nm (Fig. 4a, emission spectra from $\lambda_{ex}$ =
250 nm). In addition, a high energy narrow emission at 336 nm could
be identified in the broad PL spectra (Fig. 4b), with lifetime of -2.05 ns
(Fig. 4c). According to the previous reports[34,35], such emission could be
ascribed to free excitons, which can be captured by the lattice dis-
tortion due to strong electron-phonon interaction in metal halides,
resulting in the generation of STE.

The luminescence intensity of α-AlF₃ exhibited a linear depen-
dence on the excitation power density (more than three orders of
magnitude, Fig. 4d), indicating that the emission is originated from
photogenerated exciton (self-trapped) rather than permanent defect,
the latter of which would show saturated PL intensity upon increasing
the excitation power density[36]. Also, the emission band was not
changed upon altering the excitation power density (Supplementary
Fig. 16), further excluding the possibility of other emissive defects.
Moreover, temperature-dependent phosphorescence spectra and
cryogenic lifetime were collected (Supplementary Figs. 17, 18). The
emission intensity was enhanced upon lowering the temperature
(294 K→77 K), accompanied by a decrease in the FWHM (Fig. 4e), which
was consistent with the characteristics of electron-lattice coupling[37].
Therefore, all these above spectral features agreed well with STE.

On the other hand, although α-AlF₃ owns distorted 3D perovskite-
like structure, its room-temperature phosphorescence exhibited
interesting excitation-dependent feature. To further illustrate the
mechanism, the transitions of AlF₃ was investigated through theore-
tical calculations. Considering that the BX₆ octahedra can be

luminescent in isolated, corner-shared, and distorted structures, a
single unit of AlF₆ was calculated with the time-dependent density
functional theory (TD-DFT)[38,39]. The calculated excitation energy with
the highest oscillator strength and the emission energy from the
lowest triplet state (T₁) to the ground state (S₀) are 4.76 eV and 3.36 eV,
respectively, indicating potential large Stokes shift. Next, the natural
transition orbitals (NTO) were analyzed with Multiwfn[40]. As shown in
Fig. 5a and Supplementary Fig. 29, the transition with highest oscillator
strength happened from the un-bonding $n$ electron of F to the anti-
bonding orbitals composed by the $s$ orbital of Al and the $p$ orbital of F.
Meanwhile, such transition exhibited a typical $n{\rightarrow}\sigma^*$ character, which is
consistent with the deep UV absorption of AlF₃. For phosphorescence
transition (T₁→S₀), the frontier orbitals comprise F 2$p$ (HOMO) as well
as Al 3$p$ and F 2$p$ (LUMO). According to the selection rule for electronic
spectra[41], the electron transition of $p$-$p$ (similar to $f$-$f$ transition of lan-
thanides) orbitals is parity-forbidden, which is essential for the long-
lived phosphorescence of AlF₃. For the other BX₆ octahedra, similar
transitions ($n{\rightarrow}\sigma^*$) could also be identified and their T₁→S₀ transitions
agreed well with experimental results (Supplementary Figs. 32–36).

Recently, a number of excitation-dependent color-tunable phos-
phors featuring $n{\rightarrow}\sigma^*$ transitions were reported, in which weak
through-space interaction (TSI) was identified from the rich $n$ elec-
trons of the heteroatoms in the phosphors (cluster-induced emission,
CIE)[22–24]. Structurally, α-AlF₃ is a nonconjugated system without
through-bond conjugation, but F atoms with $n$ electrons are abundant
to form corner sharing networks (Fig. 5b). Importantly, the distances
between the adjacent F atoms in and between the AlF₆ octahedra
(e.g., intra/inter: -2.544 and -2.549 Å; inter: -3.052 Å) generally fall in
the van der Waals radii of F atom (1.47 Å)[42]. Therefore, there is possible

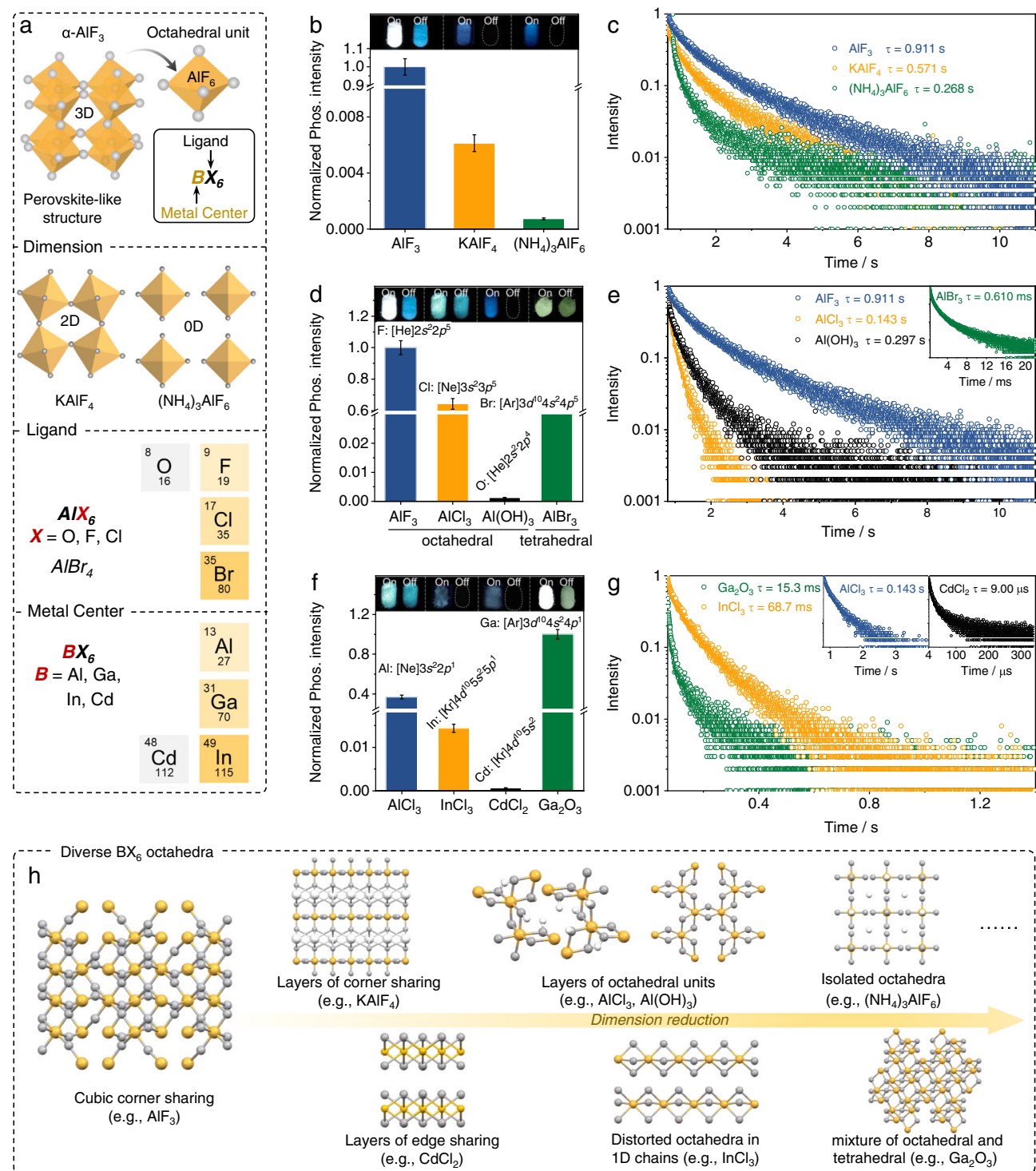

**Fig. 3 | Investigation on the $BX_6$ octahedral luminescent unit. a**, schematic diagram of chemical structure of $AlF_3$ and engineering of the octahedral basic unit; **b** and **c**, relative phosphorescence intensity and lifetime decay profiles ($\lambda_{ex} = 280$ nm) of the phosphorescence emission at 456, 480, 446 nm of $AlF_3$, $KAlF_4$, and $(NH_4)_3AlF_6$, respectively; **d**, **e**, relative phosphorescence intensity and lifetime decay profiles ($\lambda_{ex} = 280$ nm) of the phosphorescence emission at 456, 486, 450, 496 nm of $AlF_3$, $AlCl_3$, $Al(OH)_3$, and $AlBr_3$, respectively; **f**, **g**, relative phosphorescence intensity, and lifetime decay profiles ($\lambda_{ex} = 280$ nm) of the phosphorescence emission at 486, 468, 460, 500 nm of $AlCl_3$, $InCl_3$, $CdCl_2$, and $Ga_2O_3$, respectively; and **h**, summary of the diverse crystalline structures of the luminescent comprising $BX_6$ octahedra. Error bars represent standard deviation ($n = 3$).

van der Waals F···F interaction, thus leading to effective electron cloud overlap of the n electrons for $n \rightarrow \sigma^*$ transition[23]. It should be noted that F atoms also exist in NaF and KF, but no such phosphorescence could be found (Supplementary Fig. 24), further highlighting the importance of $BX_6$ octahedron.

Spectrally, the excitation spectra of α-$AlF_3$ contains two peaks, the short wavelength of which matches with its absorption, while those at longer wavelengths red shifted upon increasing the emission wavelengths (Fig. 5c). Besides, the excitation peaks at wavelength longer than the absorption were generally not detectable as a

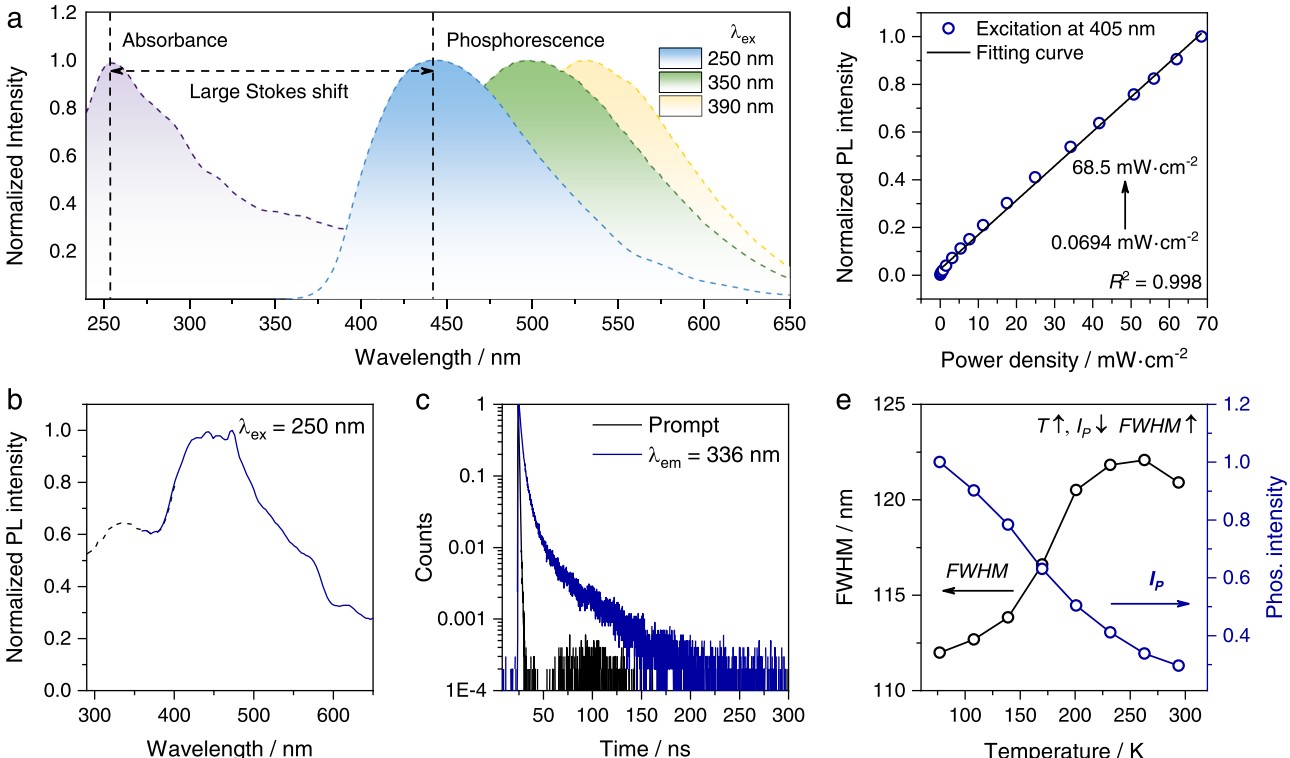

**Fig. 4 | Investigation of the STE emission of AlF₃. a** Absorbance and phosphorescence spectra ($\lambda_{ex}$ = 280, 350, and 390 nm, respectively.); **b** Normalized PL spectra ($\lambda_{ex}$ = 250 nm, the dotted line was measured without long pass filters, while the dashed line with 341 nm long pass filter placed at the emission exit); **c** lifetime of AlF₃ monitored at 336 nm ($\lambda_{ex}$ = 280 nm); **d** plot of PL intensity as a function of excitation power density ($\lambda_{ex}$ = 405 nm); **e** plots of FWHM and intensity of phosphorescence as a function of temperature ($\lambda_{ex}$ = 280 nm; delay time: 40 ms).

significant feature in the absorption spectra (Fig. 3a), which is also a typical sign of CTE. Moreover, as further calculated with the quantum mechanics and molecular mechanics (QM/MM) method, the energy gap decreased as increasing the number of AlF₆ octahedra (increasing the sizes of the clusters, Supplementary Fig. 30). Therefore, different clustering states may result in varied energy gaps (Fig. 5d), thereby excitation-dependent and color-tunable phosphorescence.

The weak interaction between AlF₆ octahedra in α-AlF₃ was further investigated with Atoms in Molecules (AIM) analysis[43]. As shown in Supplementary Fig. 31, in addition to traditional bond path in the octahedron (Al-F), there is also plenty of through-space interaction path (F···F interaction) as the AlF₆ octahedra system extended. Experimentally, upon high pressure treatment (900 MPa) to strengthen the weak intermolecular Van-der Waals interaction[44], the emission brightness of α-AlF₃ was increased somewhat (Fig. 5e, $\Phi_P$ increase from ~4.22% to ~5.46%), while the multicolor emission profiles of α-AlF₃ were not disturbed (Supplementary Fig. 19). Therefore, such interactions are expected to facilitate electron communications between the n electrons of F atoms and rigidify the system for efficient phosphorescence.

On the basis of the above analysis, similar excitation-dependent but decreased phosphorescence intensity from KAlF₄ and (NH₄)₃AlF₆ (and also other n electrons rich compounds featuring BX₆ octahedra, Fig. 3h) can thus be expected. Lowering the dimension of the AlF₆ octahedra from 3D corner-sharing (α-AlF₃) to layered (KAlF₄) and isolated ((NH₄)₃AlF₆) would decrease the inter-octahedra F···F interaction, thus weakening the through-space interaction (Supplementary Fig. 21). Moreover, the 3D corner-sharing structure may also rigidify the AlF₆ octahedra, which is beneficial for stabilization of the excited triplet states. Therefore, compared with α-AlF₃, lowering the dimension of the AlF₆ octahedra would result in sharply decreased

phosphorescence intensity and lifetime in KAlF₄ and (NH₄)₃AlF₆. For other BX₆ octahedron-containing materials, different phosphorescence performances would also be expected, due to their differences in B, X, and connection of the octahedra (Fig. 3h).

### RTP of α-AlF₃ for UV wavelength detection

Considering that the long-lived phosphorescence from AlF₃ is color-tunable in the visible range and excitation-dependent (particularly in the UV range), it was simply and conveniently explored for ultraviolet wavelength detection in a testing paper manner. As demonstrated in Fig. 6a, after mixed with Aloe vera gel for fixing, AlF₃ was coated on the filter paper. Upon UV irradiation, white luminescence from AlF₃ was excited (Fig. 6b). After ceasing of UV excitation, visible afterglow images could be obtained (Fig. 6c and Supplementary Movie 6). Notably, upon excited with UV light of different wavelengths, the afterglow emission varied from blue to orange, which could be further compared with standard color chart to confirm UV excitation wavelength (Fig. 6d and Fig. 6e). Such application provided a method for unknown UV wavelength detection and offered rapid and simple standard screening and testing of commercially UVA and UVB ultraviolet lamps available.

### Discussion

In this work, perovskite-like octahedral BX₆ was proposed a basic unit for luminescent inorganic materials. In this regard, we found that α-AlF₃ (constituted by vertex sharing AlF₆ octahedral unit) exhibited long-lived color-tunable phosphorescence emission, which could last up to 7 s and be observed by naked eyes. Besides, by lowering the dimension of the AlF₆ octahedron and changing B and X in the BX₆ octahedron, luminescence from KAlF₄, (NH₄)₃AlF₆, AlCl₃, Al(OH)₃, Ga₂O₃, InCl₃, and CdCl₂ were also obtained. The phosphorescence of AlF₃ could also be explained with the well-accepted STE mechanism of perovskite BX₆

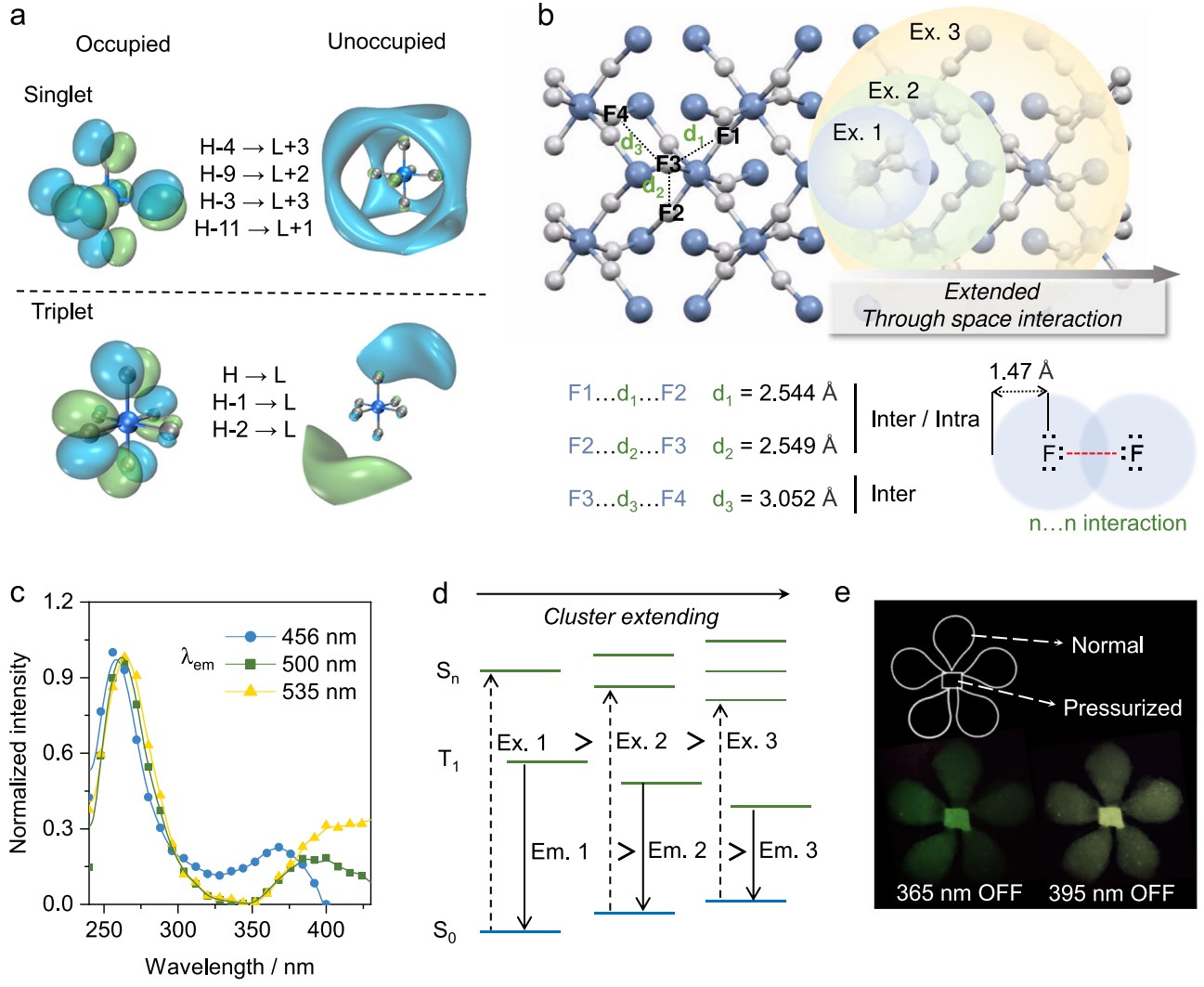

**Fig. 5 | Investigation of the phosphorescence of AlF₃. a** the TD-DFT calculated isosurfaces of occupied and unoccupied orbitals of excited singlet state with maximum oscillator strength and lowest triplet excited state in $AlF_6$ octahedral unit. **b** structural analysis of α-AlF₃ (crystalline structure from ISCD 68826). **c** excitation spectra ($\lambda_{em}$ = 456, 500, and 535 nm, respectively.) of α-AlF₃ **d** schematic energy level diagram of AlF₃ from one octahedron to network. **e** Photograph of AlF₃ afterglow with a flower pattern, consisting of untreated and pressure-treated α-AlF₃, respectively.

octahedra, together with the clusterization-triggered emission of *n* electron-rich phosphors. Therefore, $BX_6$ octahedron may be a universal structure motif for inorganic luminescent materials (Fig. 7).

For a long time, it has been well-accepted that there are core structures for organic luminescent materials, and their general photophysical properties can thus be interpreted. For inorganic luminescent materials, such core structure remains elusive. Our results here indicated that the boundary between the above two may be somewhat vague. Inorganic AlF₃ (also AlCl₃, Ga₂O₃, and etc.), featured with perovskite-like octahedral $BX_6$ basic unit, can emit long-lived phosphorescence very much similar to CTE from non-conjugated organic luminophores[22]. Moreover, typical $n{\to}\sigma^*$ transition was found in the single unit of $AlF_6$, which is also the typical character of CTE. Therefore, future development of luminescent materials integrating of both organic and inorganic luminescence mechanisms is expected to be appealing, particularly in MOFs containing both inorganic and organic units.

Last, in the long history of the afterglow phosphor family, the IIIA group elements contributed heavily (Supplementary Table 1, 2), particularly Al- and Ga-containing materials (e.g., the well-known SrAl₂O₄:Eu²⁺-Dy³⁺[13] and ZnGa₂O₄:Cr³⁺ [45]). However, the luminescence center are mostly lanthanides or transition metal ions[46] (e.g., Mn²⁺, and Cr³⁺). Here, Al- and Ga-containing afterglow phosphors with ~4% QY (e.g., AlF₃ and Ga₂O₃) are discovered. Particularly, the luminescence is come from the Al- and Ga-involved octahedral unit, not the dopants. Therefore, there is still much room for the exciting IIIA group chemistry in luminescent materials design.

## Methods
### Preparation of AlF₃
AlF₃ was prepared through calcination of AlF₃·3H₂O (99.9%, Macklin) at 350 °C for 3 h. To eliminate potential organic impurities, other AlF₃ samples were also subjected to the same calcination process as above. In addition, direct synthesis of AlF₃ was carried out through exposing aluminum metal (Aladdin, 99.999%, the highest purity available) to HF vapor from hydrofluoric acid (Macklin, 49wt. % in H₂O, 99.99998%) for 3 h, followed by calcination at 350 °C for 3 h.

### Phosphorescence measurements
Phosphorescence spectra were collected on HORIBA FluoroMax-4P with a delay time of 40 ms. For the temperature-dependent emission spectra, a model Optistat CF2 liquid nitrogen chamber

(Oxford Instruments) was used and coupled with the FluoroMax-4P spectrofluorometer. Phosphorescence lifetime and the time-resolved emission spectra (TRES) were collected on HORIBA FluoroLog-3 spectrofluorometer with 280, 355, and 390 nm spectraLED as the excitation sources, respectively. The absorption spectra of solid sample were measured on Shimadzu UV-3600 with an integrating sphere unit. The phosphorescence quantum yield ($\Phi_p$) of samples were measured in an integrating sphere (IS80, Labsphere) using 2-fluorophenylboronic acid ($\Phi_p = 0.98\%$[47]) as the reference [48].

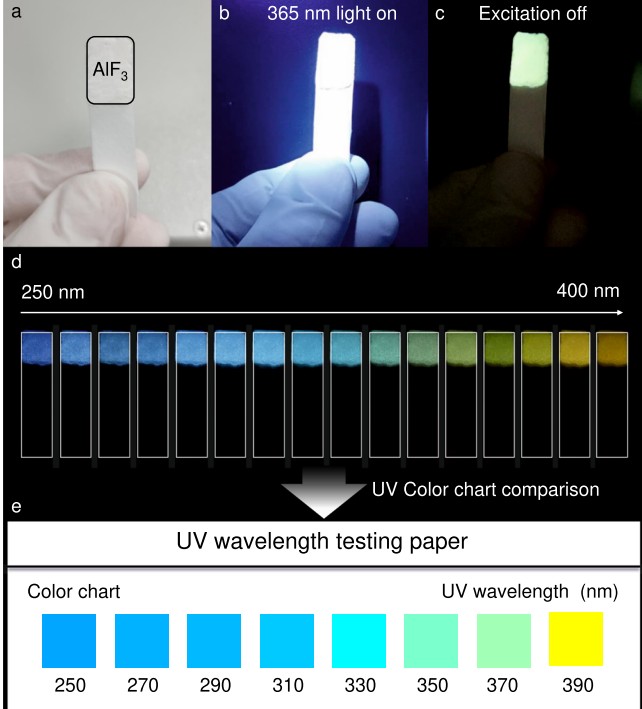

**Fig. 6 | UV wavelength detection with AlF₃-loaded testing paper. a** illustration of the testing paper coated with AlF₃; **b** a beam of UV excitation irradiated on the testing paper; **c** AlF₃-testing paper phosphorescence demonstration after ceasing 365 nm excitation; **d** multicolor phosphorescence from the AlF₃-tesing paper after ceasing a series of UV excitation; **e** color chart of UV wavelength testing paper.

## Afterglow images
The samples were excited with light selected from the Xe lamp in the Fluolog-3 spectrofluorometer (250 to 510 nm)[48]. The shutting of the excitation was controlled by instrumental software. The camera started to record video for 10 s after the excitation was turned off immediately.

## Theoretical calculation
Theoretical calculations for the octahedral unit were performed on Orca program package (Revision 4.1.1). The ground states ($S_0$) were fully optimized by M062X with ma-def2-TZVP basis set. The excitation energies in the n-th singlet ($S_n$) and n-th triplet ($T_n$) states were obtained using the time-dependent density functional theory (TD-DFT) method based on an optimized molecular structure. In order to explore detailed excited state properties, NTO (Natural transition orbitals) analysis and hole-electron analysis method were further employed using Multiwfn program. For different cluster sizes of the AlF₆ octahedra, QM/MM method was applied to obtain optimized structure of isolated, layered and cubic corner sharing AlF₆ octahedra performed in the Gaussian 09 package[49]. The central AlF₆ octahedra were treated at the (TD) M062X/6-31G+ (d,p) level, while the surrounding molecules were treated with universal force field (UFF).

For the weak interaction between AlF₆ octahedra, AIM analysis was carried out, which dictates the form of atoms in molecules and analyzes the topology of electron density. Typically, the results could be characterized by "critical points" of, namely bond (BCP), ring (RCP) and cage (CCP) critical points, representing the extreme points of electron density on the bond paths, centers of rings, and enclosed space formed by rings, respectively. It was further employed using Multiwfn program. The molecular orbitals (MO) and AIM models were all displayed using VMD.

## UV wavelength testing paper
α-AlF₃ was mixed with Aloe vera gel with mass ratio of ~1: 1, followed by fixing on the filter paper to make the mixture evenly distributed and drying. For UV wavelength detection, the testing paper was first subjected to UV excitation for 5 seconds, then the afterglow images were either naked eye observed or photo taken with a camera.

## Data availability
The data supporting the findings of this study are available within the paper and the Supplementary Information. Source data are provided with this paper.

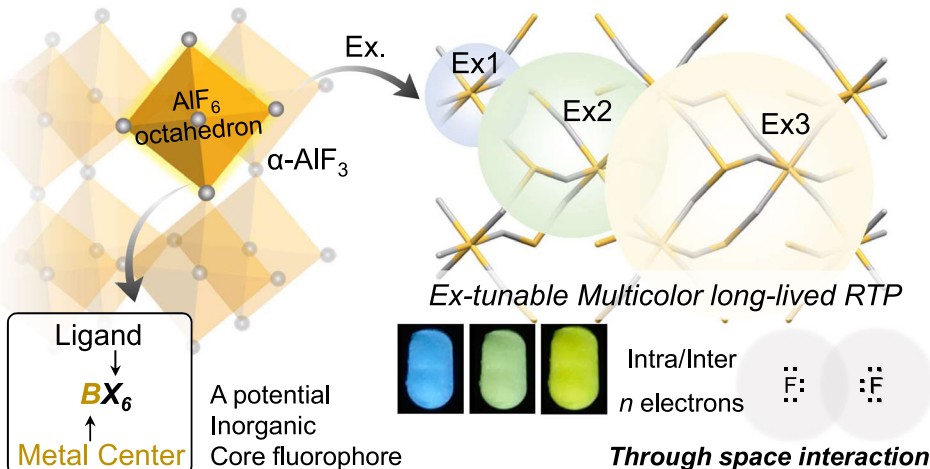

**Fig. 7 | Summary on the BX₆ octahedra-based luminescence.** Here, the luminescence mechanism of perovskite-like α-AlF₃ (and also others) was ascribed to clusterization-triggered emission of *n* electron-rich AlF₆ octahedrons, thus exhibiting excitation-tunable multicolor long-lived RTP.

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

## Acknowledgements
The authors gratefully acknowledge the financial support from the National Natural Science Foundation of China (No. 21522505) and Sichuan Science and Technology Program (No. 2021YFH0124). Detailed characterizations supported by the public Platform of Analytical & Testing Center, Sichuan University, are greatly appreciated.

## Author contributions
P.W. supervised this work. P.C. and H.Z. carried out the experimental and theoretical investigations. P.C. and P.W. wrote the manuscript.

## Competing interests
The authors declare no competing interests.
