## [Peer Review File · Nature Communications]

Multicolor Ultralong Phosphorescence from Perovskite-Like Octahedral α -AlF₃Reviewers' comments:

Reviewer #1 (Remarks to the Author):

In this work, Cao et al. explore α -AlF₃ and other compounds to propose the BX₆ octahedron as a core structure for luminescent inorganic materials. The work is a compendium of materials synthesis, characterisation, spectroscopy and theory that, although showing some promise, lack the rigour required to be published in its present form. In general, the work has several inconsistencies that I list below and does not reach the standards to be considered for publication in Nature Communications.

1) The choices of material under investigation seem arbitrary. The connection of Al-based materials with Ga₂O₃, InCl₃ and CdCl₂ is not clear. Authors here are reporting crystal structure with octahedrally coordinated metals surrounded by ligands in a variety of crystal structures. AlF₃ being the typical ReO₃ type cubic corner sharing octahedral 3D connected structure, KAIF₄ is alternating layers of corner sharing AlF₆ octahedra and BCC coordinated K. (NH₄)₃AlF₆ are isolated AlF₆ octahedra surrounded by NH₃ ligands, Al(OH)₃ are 2d layers of Al(OH)₆ octahedra hydrogen bonded to one another. Ga₂O₃ has Ga occupying a mixture of octahedral and tetrahedral sites so bucks the trend of the others. Regarding InCl₃, the In atoms are coordinated by 6 Cl atoms but they are not octahedrally arranged and form 1D chains. Finally, CdCl₂ is 2d sheets of edge sharing rather than corner sharing octahedra. A wide array of structures that are tenuously linked with the metal ligand 6 framework.

2) I disagree with the sentence on line 4, page 4. A metal halide perovskite is intrinsically a 3D structure as nicely discussed here: <https://doi.org/10.1002/aenm.201802366> and <https://doi.org/10.1021/acscenergylett.0c00039>. Please amend accordingly.

3) The way the authors explain the emission colour change with varying excitation wavelength is unsatisfactory. The reasoning that the emission is coming from STEs or multiple emission centres needs to be further backed up with experiments.

4) On line 8, page 7, it is claimed that "lowering the dimension of the AlF₆ octahedron resulted in higher quantum confinement effect and broader band gap, which decreased the possibility of photo-generated excitons (the same excitation)." I do not necessarily agree with this since the more confined the system is, the higher the probability of radiative recombination. How defective are their systems? An estimation of the defect density of each of their compounds is needed to understand the changes in PLQE observed.

5) In the previous sentence, authors claim their systems are excitonic. How do they know this?

6) In addition, I do not understand what the authors mean with "decreased the possibility of photo-generated excitons (the same excitation)" and with "the connectivity of the octahedral unit was also decreased to inhibit the flux of excitons." Could they elaborate on this?

7) The argument that lifetime follows 3d>2d>0d is weak. There is no clear connection between these structures (A-site cations different, what are the bandgap determining orbitals derived from?)

8) In the next sentence, "order and compact connection in higher dimensions favored stabilisation of the excited triplet states and thus longer lifetime". This is highly speculative, and the authors do not show evidence.

9) On line 10, page 8, authors claim that "The lifetime of AlCl₃ is shortened as compared with AlF₃ (Fig. 3e), probably because of the larger atomic number of Cl as compared with F (heavy atom effect).25-26" This needs to be commented on further since the two reviews mention only very heavy elements such as thallium and iodide. Additionally, the heavy atom effect is usually used to promote phosphorescence. The lack of phosphorescence yields for these compounds makes them harder to compare.

10) The following sentences are wrong: "The luminescence of most MHPs originates from the self-trapped exciton (STE) emission, in which the exciton are captured by the lattice distortion due to strong electron-phonon interaction" and "The well-accepted luminescence mechanism of Bx₆ octahedron-based MHPs (self-trapped excton, STE)." Actually, emission in most of MHPs does not rely on STEs.

11) On line 5, page 11, authors write "considering that extracting basic octahedral unit from the extended network, which could be described by ligand field theory and molecular orbital (MO) model, were useful for explaining the transitions that occur between the states to identify their allowed or forbidden properties." Is the suggestion here that the carriers are so isolated as STEs that this

structure is effectively like 0D isolated octahedra even though they are all well connected in a 3D lattice? If this is the case, why doesn't the 2D or 0d analogues of the AIF6 octahedral structure have the same emission characteristics as the 3D equivalent?

12) Line 16, page 13: "Therefore, the AIF6 vertex shared 3D structures would result in different luminescence centers and exciton motion, thus different excitation and multicolor phosphorescence" Why is it appropriate then to model them with a single octahedral unit?

Other comments:

- The electronic structure of Al is $3s^2 3p^1$, the $3d^0$ should be omitted.
- Could the authors show the phosphorescence spectra a longer wavelengths? In Figure 1 and 2 (and more) of the supporting information it is not clear to this referee why the authors cut the x-axis at 550 nm.
- Generally, decay times and amplitudes are missing (fits missing).
- On line 9, page 3, please rewrite this sentence: "While for inorganic luminescent materials, although structurally diverse and a largely number of host materials for doping of transition metal ions (e.g., ZnS12 and SrAl2O413), no core structure as the light-emitting unit reported currently."
- On line 16, page 3, please note halide perovskites are not a new type of semiconductors. They have been around for decades.
- What do the authors mean with "smaller sized AIF6 connection" (vs "larger sized AIF6 connection")?
- Figure 2c: lifetime decay for which wavelength?
- Figure 3a: 3d vs 2d vs 0D not very well illustrated. 0D could be 3D if properly connected.
- SI Figure 3: what do the PDF files correspond to?
- SI Figure 14: KAIF4 doesn't seem to be phase-pure.

Reviewer #2 (Remarks to the Author):

This work from Wu and coworkers might have some potential to be featured as a fundamental study in Nature Communications. The topic is strictly basic science and the attempt to tackle fundamental questions is clear.

However, I have some major concerns which hinder my total approval and therefore I suggest major revisions.

1. The language is often difficult to follow. This lack of clear expression (with some real grammatical mistakes) might compromise the clear understanding of what the authors want to communicate to the reader. This is surely a very relevant drawback.
2. How do the authors justify the excitation dependent luminescence in terms of orbital diagram? It is not clear how they explain change in color of the luminescence by changing excitation wavelength.
2. Do the author exclude completely a contribution to the long-lived luminescence given by thermally activated delayed fluorescence? The increase of emission intensity with lowering of the temperature makes believe that TADF is not existing in this system, but still it cannot be ruled out just based on T-dependent measurements of the PL and theoretical calculations. The effective existence of the triplet as the emissive species might be seen by other time-resolved techniques like transient absorption/pump-probe spectroscopy or even EPR. Can the author present an argument against the need to carry out also similar more detailed investigations to really explain the observed phenomena?
3. The emission in metal halide perovskite is not always due to the STE. In 2D perovskite both excitonic emission and STE can be observed. In 0D it is most of the time purely excitonic. Why do the authors claim that STE is the typical emissive feature in MHPs?
4. I do not understand the chemiluminescence measurements. What do the authors want to show with these data?
5. It is not clear why the formation of the triplet should be favored in these systems. There is only one very sibylline sentence that should explain it, but it is indeed, sibylline to me (The excited electron could be stabilized at T1 state due to orbital parity rule, thus exhibiting effective long-lived phosphorescence, which could be ascribed to the p-p parity forbidden transition). The authors should better clarify why the formation of the triplet is so favored in these systems. In inorganics it is normally the heavy atom effect that drives its formation. Here anyway we are dealing with light elements. The authors may understand that it is not so logical based on generally assumed scientific knowledge to guess that a triplet should form in similar materials. Can the author bring other

examples from other inorganic materials forming the triplet and being constituted of light elements? In MHP the PL lifetimes are typically those of singlet states, for example.

6. The authors claim that in organics there are certain moieties that are typically luminescent. I think this is a very arguable statement. It is true that many organic semiconductors featuring such moieties are highly luminescent, but it is also the tendency to aggregate that drives the luminescence there (the more they can self-aggregate, the more the PL can be quenched. therefore the type of substituents play a major role there). In the end similar affirmations are very relevant to introduce the uniqueness of the fundamental question that they want to raise and to which they want to give an unambiguous answer.

Manuscript ID: NCOMMS-21-37768

Title: Multicolor Ultralong Phosphorescence from Perovskite-Like Octahedral α -AlF₃

We thank the two reviewers for their efforts in reviewing our manuscript and providing valuable and constructive comments which have greatly improved both its presentation and scientific content. All comments and suggested changes were carefully considered and addressed during revision of the original manuscript that we now resubmit to *Nature Communication* for consideration.

Since the MS was thoroughly revised, it is difficult to accurately indicate the exact places where the changes were made in the revised MS.

General comment regarding reorganisation and focus of the revised manuscript:

Encouraged by both the helpful and critical comments of the two reviewers, we now have established the mechanism of self-trapped emission of perovskite-like α -AlF₃ with additional free exciton emission (presented in Figure 3 and Figure 4). Moreover, the excitation dependent color tunable phosphorescence from AlF₃ was ascribed to cluster-trigger emission (presented in Figure 5). The AlF₆ octahedron brought close proximity of F atoms, resulting in van der Waals F...F interaction and through-space conjugation of the n electrons for $n \rightarrow \sigma^*$ transition. Such mechanism was confirmed with both positive (e.g., KAlF₄ and (NH₄)₃AlF₆, different octahedral connections) and negative (NaF and KF, no octahedra) proofs.

Overall, a completely revised manuscript is presented which includes additional data and highlights the mechanism of STE and CTE. We think our work may be fundamentally important for both inorganics and organics:

1. In an inorganic material with light elements, exciting phosphorescence was discovered;
2. Typically used mechanism in organics, CTE, was found in an inorganic material AlF₃.

We also discuss how these findings expand our understanding of the BX₆ octahedral from the structural perspective (Figure 3). We believe that with the substantial manuscript revision, including its reorganisation, our message and novelty of key observations have become much clearer and also addressed the concerns of the referees.

Reviewer #1

Remarks to the author:

In this work, Cao et al. explore α -AlF₃ and other compounds to propose the BX₆ octahedron as a core structure for luminescent inorganic materials. The work is a compendium of materials synthesis, characterisation, spectroscopy and theory that, although showing some promise, lack the rigour required to be published in its present form. In general, the work has several inconsistencies that I list below and does not reach the standards to be considered for publication in Nature Communications.

Reply: Thanks for your valuable comments!

The MS was thoroughly revised according to the suggestions from two reviewers. Just as the other reviewer stated, it is normally the heavy atom effect that drives the formation of triplet in most inorganics. Here, α -AlF₃ contains light elements. Therefore, the phenomena here is truly interesting. The excitation-dependent color tunable phosphorescence was explained with clusterization-triggered emission (CTE), in which the octahedra brought close proximity of F atoms for weak through-space conjugation.

Generally, the other reviewer held similar opinion as you that the work is interesting and supported it to be published at this journal. We do hope the reversions are acceptable for you to support publication of this MS.

Comment 1. The choices of material under investigation seem arbitrary. The connection of Al-based materials with Ga₂O₃, InCl₃ and CdCl₂ is not clear. Authors here are reporting crystal structure with octahedrally coordinated metals surrounded by ligands in a variety of crystal structures. AlF₃ being the typical ReO₃ type cubic corner sharing octahedral 3D connected structure, KAlF₄ is alternating layers of corner sharing AlF₆ octahedra and BCC coordinated K. (NH₄)₃AlF₆ are isolated AlF₆ octahedra surrounded by NH₃ ligands, Al(OH)₃ are 2d layers of Al(OH)₆ octahedra hydrogen bonded to one another. Ga₂O₃ has Ga occupying a mixture of octahedral and tetrahedral sites so bucks the trend of the others. Regarding InCl₃, the In atoms are coordinated by 6 Cl atoms but they are not octahedrally arranged and form 1D chains.

Finally, CdCl_2 is 2d sheets of edge sharing rather than corner sharing octahedra. A wide array of structures that are tenuously linked with the metal ligand 6 framework.

Reply: Thanks very much for your expert comments!

We must express our sincere appreciation for you on the above important structural information, which was absent in the original version of our MS. For InCl_3 , although not octahedrally arranged, it can be regarded as distorted BX_6 octahedra in the 1D chains. Therefore, although structurally diverse, all these materials possess the same BX_6 octahedra unit (B: the central cation; X: the ligand). Most importantly, they all exhibited RTP, thus confirming our initial envision that the BX_6 octahedra may be a core structure for luminescent inorganic materials.

In the revised version, the above structural information was added in Figure 2 for further validation of the initial envision.

Figure R1 Investigation on the BX_6 octahedral luminescent unit: a, schematic diagram of chemical structure of AlF_3 and engineering of the octahedral basic unit; b and c, relative phosphorescence intensity and lifetime decay profiles ($\lambda_{\text{ex}} = 280$ nm) of the phosphorescence emission at 456, 480, 446 nm of AlF_3 , KAIF_4 , and $(\text{NH}_4)_3\text{AlF}_6$, respectively; d and e, relative phosphorescence intensity and lifetime decay profiles ($\lambda_{\text{ex}} = 280$ nm) of the phosphorescence emission at 456, 486, 450, 496 nm of AlF_3 , AlCl_3 , $\text{Al}(\text{OH})_3$, and AlBr_3 , respectively; f and g, relative phosphorescence intensity and lifetime decay profiles ($\lambda_{\text{ex}} = 280$ nm) of the phosphorescence emission at 486, 468, 460, 500 nm of AlCl_3 , InCl_3 , CdCl_2 , and Ga_2O_3 , respectively; and h, summary of the diverse crystalline structures of the luminescent comprising BX_6 octahedra.

Comment 2. I disagree with the sentence on line 4, page 4. A metal halide perovskite is intrinsically a 3D structure as nicely discussed here: <https://doi.org/10.1002/aenm.201802366> and <https://doi.org/10.1021/acsenenergylett.0c00039>. Please amend accordingly.

Reply: Thanks very much for your expert comments!

To avoid potential misleading on the intrinsic 3D perovskite and its low dimensional derivatives, the related part was revised as follows:

“The core structure of luminescent MHPs can be described as BX_6 octahedron (Fig. 1), which is constituted by the central cation (B, hexa-coordinated) and six halide atoms ($X = Cl, Br, I$). Normally, the BX_6 octahedra is organized in an all-corner-sharing 3D network. Due to the adjustable octahedral connectivity, a series of lower dimensional metal halide-based luminescent perovskite derivatives have been reported. On the other hand, the central cation and ligand halides could be altered, leading to tunable luminescence performance from 3D and lower dimension metal halides.”

Comment 3. The way the authors explain the emission colour change with varying excitation wavelength is unsatisfactory. The reasoning that the emission is coming from STEs or multiple emission centres needs to be further backed up with experiments.

Reply: Thanks for the valuable suggestion!

On one hand, STE from α -AlF₃ was further confirmed with new experimental results. The related spectral features of α -AlF₃ were summarized as follows:

1. Broad-band emission with large stokes shift (> 180 nm);
2. Linear-dependence of the luminescence intensity on the excitation power density (the emission band was not changed upon altering the excitation power density as shown in Figure R2, further excluding the possibility of emissive defects);
3. Increased emission intensity but decreased FWHM when lowering temperature;
4. A high energy narrow emission at ~336 nm ($\tau \sim 2.05$ ns) could be identified in the broad PL spectra (Figure R3), which could be ascribed to free excitons (captured by the lattice distortion due to strong electron-phonon interaction in metal halides, resulting in the generation of STE).

Figure R2. (a) Excitation power-dependent PL spectra and (b) plot of PL intensity as a function of excitation power density ($\lambda_{\text{ex}} = 405 \text{ nm}$).

Figure R3. (A) Normalized PL spectra (The dotted line was measured without long pass filters while the dashed line with 341 nm long pass filter placed at the emission exit. Both of them were excited at 250 nm under the same instrumental condition); (B) lifetime of AlF_3 monitored at 336 nm (excited by 280 nm).

On the other hand, the multicolor phosphorescence of $\alpha\text{-AlF}_3$ were thoroughly re-considered. Generally, there are two possible mechanisms accounting for this unique phenomenon: 1) impurity and 2) multiple emission centers.

First, we prepared AlF_3 from by exposing aluminum metal to HF vapor (both are the highest purity available). As can be seen from Figure R4 below, almost identical emission features were obtained. Therefore, the multicolor phosphorescence was exactly from AlF_3 .

Figure R4. Normalized phosphorescence spectra and phosphorescent lifetime of original AlF_3 and prepared samples from treating aluminum metal with hydrofluoric acid excited by 280, 350 and 390 nm, respectively.

Considering the composition and structure of $\alpha\text{-AlF}_3$, the excitation-dependent color-tunable phosphorescence should be originated from the connected AlF_6 octahedra. Since the transition in the AlF_6 octahedra was identified as $n \rightarrow \sigma^*$, we attributed it to clusterization-triggered emission (CTE, for details, see Zhang, H. K., et al., Clusterization-triggered emission: Uncommon luminescence from common materials, *Mater. Today* **2020**, *32*, 275-292.). Two typical structural features could be extracted from these CTE luminophores:

- (i) Nonconjugated systems without traditional through-bond conjugation (TBC);
- (ii) Abundant heteroatoms with lone pair electrons. The close proximity of the n electrons may result in weak through space conjugation, leading to multiple emission centers of varied electron localization and thus excitation-dependent color-tunable emission (Figure R5).

Similar to N and O atoms in previous excitation-dependent color-tunable phosphors featuring $n \rightarrow \sigma^*$ transitions, F also possesses rich lone pair (n) electrons. Importantly, the distances between adjacent F atoms in and between the AlF_6 octahedra (e.g., inter/intra: ~ 2.544 and $\sim 2.549 \text{ \AA}$; intra: $\sim 3.052 \text{ \AA}$) generally fall in the van der Waals radii of F atom (1.47 \AA). Therefore, there is possible van der Waals F...F interaction, thus leading to effective through-space conjugation of the n electrons for $n \rightarrow \sigma^*$ transition.

Figure R5. Structural analysis of $\alpha\text{-AlF}_3$ (crystalline structure from ICSD 68826).

Besides the excitation-dependent color-tunable phosphorescence, two peaks could be identified in the excitation spectra of $\alpha\text{-AlF}_3$, the short wavelength of which match with its absorption, while those at longer wavelengths red shifted upon increasing the emission wavelengths (Figure R6 below). Besides, the excitation peaks at wavelength longer than the absorption were generally not detectable as a significant feature in the absorption spectra. The above spectra feature is also a typical sign of CTE (e.g., see Zhang, H. K., Tang, B. Z. Through-Space Interactions in Clusteroluminescence, *JACS Au* **2021**, *1*, 1805-1814.).

Figure R6. **a** Absorption and phosphorescence spectra; **b** Phosphorescence spectra at different excitation wavelengths; **c** Excitation spectra at different wavelengths.

To further confirm CTE, the powder of α -AlF₃ was subjected to high pressure tableting. After treatment (10 MPa), the multicolor emission profiles were not disturbed, but the emission brightness was increased somewhat (Figure R7), accompanied by Φ_P increase from ~4.22% to ~5.46%. Probably, the high pressure treatment would result in slight packing compactness of the AlF₆ octahedra, which favored the communication between the n electrons of F atoms and suppressed nonradiative deactivation.

Figure R7. photographs of AlF₃ afterglow and measured phosphorescence quantum yield ($\lambda_{ex} = 285$ nm) of original sample and 10 MPa tableting AlF₃, respectively.

Last, the BX_6 octahedra structure is important for the above CTE. On one hand, the distances between adjacent F atoms were brought close proximity for potential $F \cdots F$ interactions. On the other hand, other species containing abundant F atoms but no such octahedra structure (e.g., NaF or KF), do not show any luminescence.

On the basis of the above analysis, the discussion about STE of $\alpha\text{-AlF}_3$ and its excitation-dependent color-tunable phosphorescence was totally re-written. Please see our revised MS. Besides, we think the response here can also answer some of your comments below.

Comment 4. On line 8, page 7, it is claimed that "lowering the dimension on of the AlF_6 octahedron resulted in higher quantum confinement effect and broader band gap, which decreased the possibility of photo-generated excitons (the same excitation)." I do not necessarily agree with this since the more confined the system is, the higher the probability of radiative recombination. How defective are their systems? An estimation of the defect density of each of their compounds is needed to understand the changes in PLQE observed

Reply: Thanks very much for your valuable comment! We agree with the reviewer that the explanation in the previous MS was problematic, and the related contents were deleted accordingly.

Here, the decreased luminescence intensity was ascribed to the decreased through space conjugation. As shown in Figure R8 below, lowering the dimension of the AlF_6 octahedra from 3D corner-sharing ($\alpha\text{-AlF}_3$) to layered (KAlF_4) and isolated ($(\text{NH}_4)_3\text{AlF}_6$) would decrease the inter-octahedra $F \cdots F$ interaction, thus weakening the through-space conjugation. Moreover, the 3D corner-sharing structure may also rigidify the AlF_6 octahedra, which is beneficial for stabilization of the excited triplet states. Therefore, compared with $\alpha\text{-AlF}_3$, lowering the dimension of the AlF_6 octahedra would result in sharply decreased phosphorescence intensity in KAlF_4 and $(\text{NH}_4)_3\text{AlF}_6$.

Figure R8. Schematic illustration of AlF₆ octahedra space conjugation with diverse dimension (3D network, layers of corner sharing and isolated).

We agree with the reviewer that defects could quench the recombination of excitons and thus lower the PLQE. However, potential defect effect on the emission was ruled out (see our reply to your comment 3). Moreover, the decreased intensity and lifetime could be well explained with CTE. Therefore, we do not ascribe the decreased intensity and lifetime to defect density.

Comment 5. In the previous sentence, authors claim their systems are excitonic. How do they know this?

Reply: Thanks very much for your valuable advice!

For excitonic emission, the emission intensity typically exhibits linear dependence on the excitation power. While involving defects, it should be sublinear when the limited number of defect states become saturated (for details, see Schmidt T. et al. Excitation-power dependence of the near-band-edge photoluminescence of semiconductors. *Phys. Rev. B*, **1992**, 45, 8989-8994).

As can be seen from Figure R2 below, the luminescence intensity was linearly correlated with the excitation power for over 3 orders of magnitude.

Figure R2. (a) Excitation power-dependent PL spectra and (b) plot of PL intensity as a function of excitation power density ($\lambda_{\text{ex}} = 405$ nm).

In addition, a high energy narrow emission peak at 336 nm was observed in broadband PL spectra excited by 250 nm with a short lifetime of ~ 2.05 ns (Figure R3), which may be attributed to emission from free excitons (for details, see Yuan, Z. et al., One-dimensional organic lead halide perovskites with efficient bluish white-light emission, *Nat. Commun.* **2017**, 8, 14051). Free excitons can be captured by the lattice distortion due to strong electron-phonon interaction in metal halides, resulting in the generation of STE.

Figure R3. (A) Normalized PL spectra (The dotted line was measured without long pass filters while the dashed line with 341 nm long pass filter placed at the emission exit. Both of them were excited at 250 nm under the same instrumental condition); (B) lifetime of AlF_3 monitored at 336 nm (excited by 280 nm).

Comment 6. In addition, I do not understand what the authors mean with "decreased the possibility of photo-generated excitons (the same excitation)" and with "the connectivity of the octahedral unit was also decreased to inhibit the flux of excitons." Could they elaborate on this?

Reply: Thanks very much for your valuable advice!

As detailed in our reply to your comment 3 and 4, the related discussions were totally re-written.

Comment 7. The argument that lifetime follows $3d > 2d > 0d$ is weak. There is no clear connection between these structures (A-site cations different, what are the bandgap determining orbitals derived from?)

Reply: Thanks very much for your comments!

First, thanks for the careful reviewing, and now pure phase KAlF_4 was examined (as indicated in your last comment). The lifetime difference ($3D > 2D > 0D$) is much more appreciable than that in the previous version (Figure R8 below).

Figure R8. a Normalized PL spectra. The dotted line was measured without long pass filters while

There are two structural differences in AlF_3 (3D), KAIF_4 (2D), and $(\text{NH}_4)_3\text{AlF}_6$ (0D):

1. KAIF_4 and $(\text{NH}_4)_3\text{AlF}_6$ possess A-site cations, while AlF_3 not. Usually, A site cations contribute to lattice stabilization, but do not hybridize to the frontier molecule orbitals. In other words, they contribute little to the photophysical process (luminescence).
2. The connection of the AlF_6 octahedra was different, which influence the phosphorescence in the following two aspects: (1) F...F interaction; (2) rigidification of the AlF_6 octahedra. As indicated in our reply to your comment 4 above, multi-dimensional linkage of the AlF_6 octahedra is beneficial for the F...F interaction and can rigidify AlF_6 octahedra (similar to other nonconventional luminophores through effective heteroatoms interactions).

Therefore, the lifetime follows the order of AlF_3 (3D) > KAIF_4 (2D) > $(\text{NH}_4)_3\text{AlF}_6$ (0D).

Comment 8. In the next sentence, "order and compact connection in higher dimensions favored stabilisation of the excited triplet states and thus longer lifetime". This is highly speculative, and the authors do not show evidence.

Reply: Thanks very much for your comments!

As detailed in our reply to your comment 7, the related discussions were totally re-written.

In our system, generation of the triplet excitons were determined by following two factors:

1. Heteroatoms (F) are abundant in AlF_3 , resulting in through space conjugation of the n electrons of F atoms. Such weak conjugation eventually lead to phosphorescence from AlF_3 (similar to previous nonconventional luminophores, e.g., Zhou Q. et al. Clustering-Triggered Emission of Nonconjugated Polyacrylonitrile. *Small*, **2016**, 12, 6586-6592; Zhou Q., et al. A clustering-triggered emission strategy for tunable multicolor persistent phosphorescence. *Chem. Sci.*, **2020**, 11, 2926-293; Zhang, H.; Tang, B. Z. Through-Space Interactions in Clusteroluminescence, *JACS Au* **2021**, 1, 1805-1814).
2. The rigidification of the AlF_6 octahedra follows the order of AlF_3 (3D network) > KAlF_4 (2D, layered corner sharing) > $(\text{NH}_4)_3\text{AlF}_6$ (0D, isolated). In fact, previous references on nonconventional luminophores also exhibited similar stabilizing trends, for example, polyacrylonitrile (Zhou Q., et al. Clustering-Triggered Emission of Nonconjugated Polyacrylonitrile. *Small*, **2016**, 12, 6586-6592), xylitol (Wang Y. Z., et al. Emission and Emissive Mechanism of Nonaromatic Oxygen Clusters. *Macromol. Rapid Commun.*, **2018**, 39, 1800528) and cyanoacetic acid (Fang M., et al. Unexpected room-temperature phosphorescence from a non-aromatic, low molecular weight, pure organic molecule through the intermolecular hydrogen bond. *Mater. Chem. Front.*, **2018**, 2, 2124-2129.). In these works, various intra- or intermolecular interactions resulted in sufficiently rigidified conformations and generate room-temperature phosphorescence.

Comment 9. On line 10, page 8, authors claim that "The lifetime of AlCl_3 is shortened as compared with AlF_3 (Fig. 3e), probably because of the larger atomic number of Cl as compared with F (heavy atom effect).²⁵⁻²⁶" This needs to be commented on further since the two reviews mention only very heavy elements such as thallium and iodide. Additionally, the heavy atom effect is usually used to promote phosphorescence. The lack of phosphorescence yields for these compounds makes them harder to compare.

Reply: Thanks very much for your expert comment!

We are regretful for such a mistake. Here, the QY of AlCl_3 (~2.0%) is lower than that of AlF_3 (~4.22%). Therefore, it cannot be explained with heavy atom effect. Alternatively, we think the

difference in QY and lifetime may be explained by the following two issues: (1) different connection of the BX_6 octahedra; (2) different CTE effect from F and Cl.

The related description was revised as follows:

“Through altering halide composition from F to Br, their emission spectra are readily tunable from blue to yellow (Insets of Fig. 3d), which was similar with the emission tunable perovskite materials through halide engineering.”

Besides, when discussing the CTE effect, we also added the following explanations:

“For other BX_6 octahedron-containing materials, different phosphorescence performances would also be expected, due to their differences in B, X, and connection of the octahedra (Fig. 3h).”

Comment 10. The following sentences are wrong: "The luminescence of most MHPs originates from the self-trapped exciton (STE) emission, in which the exciton are captured by the lattice distortion due to strong electron-phonon interaction" and "The well-accepted luminescence mechanism of BX_6 octahedron-based MHPs (self-trapped exciton, STE)." Actually, emission in most of MHPs does not rely on STEs.

Reply: Thanks very much for your kind reminding!

The related description was revised as follows:

“For the intrinsic luminescence of MHPs, although defect (vacancy)-induced emission has been reported, self-trapped exciton (STE) emission is also well-accepted.”

Comment 11. On line 5, page 11, authors write "considering that extracting basic octahedral unit from the extended network, which could be described by ligand field theory and molecular orbital (MO) model, were useful for explaining the transitions that occur between the states to identify their allowed or forbidden properties." Is the suggestion here that the carriers are so isolated as STEs that this structure is effectively like 0D isolated octahedra even though they are all well connected in a 3D lattice? If this is the case, why doesn't the 2D or 0d analogues of the AlF_6 octahedral structure have the same emission characteristics as the 3D equivalent?

Reply: Thanks very much for your comments and we are sorry for such misleading.

In essence, we also thought the relationship was somewhat unreasonable. Therefore, sentence mentioned in this comment was deleted and the related part was re-written as follows:

“Considering that the BX_6 octahedra can be luminescent in isolated, corner-shared, and distorted structures, a single unit of AlF_6 was calculated with the time-dependent density functional theory (TD-DFT).”

Comment 12. Line 16, page 13: "Therefore, the AlF_6 vertex shared 3D structures would result in different luminescence centers and exciton motion, thus different excitation and multicolor phosphorescence" Why is it appropriate then to model them with a single octahedral unit?

Reply: Thanks very much for your comments!

As detailed in our reply to your comment 3, the related discussions were totally re-written.

In this work, the excitation-dependent color-tunable phosphorescence of AlF_3 was ascribed to CTE. The related feature was thus studied with first a single octahedral unit and then network. In addition, similar idea was also found in graphene oxide (GO) and its excitation-dependent emission was ascribed to the size effect of sp^2 clusters (Figure R9) [*Adv. Opt. Mater.*, **2013**, 1, 926-932].

Figure R9. Schematic illustration of mechanism accounting for graphene oxide (reference) and AlF_3 (this work) excitation-dependent luminescence.

Minors:

1) The electronic structure of Al is $3s^23p^1$, the $3d0$ should be omitted.

Reply: Thanks, and it was revised accordingly.

2) Could the authors show the phosphorescence spectra a longer wavelengths? In Figure 1 and 2 (and more) of the supporting information it is not clear to this referee why the authors cut the x-axis at 550 nm.

Reply: Thanks! Such actions were not reasoned. To avoid potential confusion, all the figures about phosphorescence were changed with x-axis up to 650 nm.

3) Generally, decay times and amplitudes are missing (fits missing).

Reply: Thanks!

The method for the fitting was given the Supporting Information. In addition, the results of the fitting (average lifetime) was added in Figure 3.

4) On line 9, page 3, please rewrite this sentence: "While for inorganic luminescent materials, although structurally diverse and a largely number of host materials for doping of transition metal ions (e.g., ZnS and SrAl₂O₄), no core structure as the light-emitting unit reported currently."

Reply: Thanks very much for your comments! The revised sentence is listed below:

“While for inorganic luminescent materials, although structurally diverse and mostly acting as host materials for doping of transition- or rare-earth metal ions (e.g., ZnS and SrAl₂O₄), core structure as the light-emitting unit has seldom been reported and explored like organics”

5) On line 16, page 3, please note halide perovskites are not a new type of semiconductors. They have been around for decades.

Reply: Thanks! The revised description about halide perovskites was added in our manuscript as follows:

“All-inorganic metal halide perovskites (MHPs), a type of semiconductor materials with excellent photoelectric properties, have been widely used in solar cells, LED, and thermoelectric modules.”

6) What do the authors mean with "smaller sized AlF_6 connection" (vs "larger sized AlF_6 connection")?

Reply: Thanks very much for your comment!

As detailed in our response to your comment 3, the excitation-dependent luminescence was ascribed to CTE. The phrase “smaller sized AlF_6 connection” was replaced with different scale of through space conjugation.

- Figure 2c: lifetime decay for which wavelength?

Reply: Thanks! The detailed information for lifetime collection was added in Figure 2c.

- Figure 3a: 3d vs 2d vs 0D not very well illustrated. 0D could be 3D if properly connected.

Reply: Thanks very much for your valuable comments!

The illustration about 0D was revised to avoid potential misleading.

- SI Figure 3: what do the PDF files correspond to?

Reply: Thanks very much for your comments!

The PDF files correspond to AlF_3 with space group of $R3c$ and $Cmcm$. Such information was added in the caption as suggested.

- SI Figure 14: KAlF_4 doesn't seem to be phase-pure.

Reply: Thanks very much for your careful and expert reviewing.

First, now pure phase KAlF_4 was examined (Figure R10 below), and the excitation-dependent phosphorescence spectra, lifetime, and afterglow images were all replaced (all the information was similar to those in our last version).

Most importantly, the use of pure phase KAlF_4 led to more appreciable lifetime difference ($3D >$

2D > 0D) as compared with the results in our previous version.

Figure R10. XRD pattern of KAIF_4

Reviewer 2

Remarks to the author:

This work from Wu and coworkers might have some potential to be featured as a fundamental study in Nature Communications. The topic is strictly basic science and the attempt to tackle fundamental questions is clear. However, I have some major concerns which hinder my total approval and therefore I suggest major revisions.

Reply: Thanks very much for your positive comments and professional suggestions on our work.

Comment 1. The language is often difficult to follow. This lack of clear expression (with some real grammatical mistakes) might compromise the clear understanding of what the authors want to communicate to the reader. This is surely a very relevant drawback.

Reply: Thanks! The language was polished during the reversion of this MS.

Comment 2. How do the authors justify the excitation dependent luminescence in terms of orbital diagram? It is not clear how they explain change in color of the luminescence by changing excitation wavelength.

Reply: Thanks very much for your expert comment! In fact, similar concern was also raised by Reviewer I.

The multicolor phosphorescence of α -AlF₃ were thoroughly re-considered. Generally, there are two possible mechanisms accounting for this unique phenomenon: 1) impurity and 2) multiple emission centers.

First, we prepared AlF₃ from by exposing aluminum metal to HF vapor (both are the highest purity available). As can be seen from Figure R4 below, almost identical emission features were obtained. Therefore, the multicolor phosphorescence was exactly from AlF₃.

Figure R4. Normalized phosphorescence spectra and phosphorescent lifetime of original AlF_3 and prepared samples from treating aluminum metal with hydrofluoric acid excited by 280, 350 and 390 nm, respectively.

Considering the composition and structure of $\alpha\text{-AlF}_3$, the excitation-dependent color-tunable phosphorescence should be originated from the connected AlF_6 octahedra. Since the transition in the AlF_6 octahedra was identified as $n \rightarrow \sigma^*$, we attributed it to clusterization-triggered emission (CTE, for details, see Zhang, H. K., et al., Clusterization-triggered emission: Uncommon luminescence from common materials, *Mater. Today* **2020**, *32*, 275-292.). Two typical structural features could be extracted from these CTE luminophores:

- (iii) Nonconjugated systems without traditional through-bond conjugation (TBC);
- (iv) Abundant heteroatoms with lone pair electrons. The close proximity of the n electrons may result in weak through space conjugation, leading to multiple emission centers of varied electron localization and thus excitation-dependent color-tunable emission (Figure R5).

Similar to N and O atoms in previous excitation-dependent color-tunable phosphors featuring $n \rightarrow \sigma^*$ transitions, F also possesses rich lone pair (n) electrons. Importantly, the distances between adjacent F atoms in and between the AlF_6 octahedra (e.g., inter/intra: ~ 2.544 and $\sim 2.549 \text{ \AA}$; intra: $\sim 3.052 \text{ \AA}$) generally fall in the van der Waals radii of F atom (1.47 \AA). Therefore, there is possible van der Waals F...F interaction, thus leading to effective through-space conjugation of the n electrons for $n \rightarrow \sigma^*$ transition.

Figure R5. Structural analysis of $\alpha\text{-AlF}_3$ (crystalline structure from ICSD 68826).

Besides the excitation-dependent color-tunable phosphorescence, two peaks could be identified in the excitation spectra of $\alpha\text{-AlF}_3$, the short wavelength of which match with its absorption, while those at longer wavelengths red shifted upon increasing the emission wavelengths (Figure R6 below). Besides, the excitation peaks at wavelength longer than the absorption were generally not detectable as a significant feature in the absorption spectra. The above spectra feature is also a typical sign of CTE (e.g., see Zhang, H. K.; Tang, B. Z. Through-Space Interactions in Clusteroluminescence, *JACS Au* **2021**, *1*, 1805-1814.).

Figure R6. **a** Absorption and phosphorescence spectra; **b** Phosphorescence spectra at different excitation wavelengths; **c** Excitation spectra at different wavelengths.

To further confirm CTE, the powder of α - AlF_3 was subjected to high pressure tableting. After treatment (10 MPa), the multicolor emission profiles were not disturbed, but the emission brightness was increased somewhat (Figure R7), accompanied by Φ_{P} increase from $\sim 4.22\%$ to $\sim 5.46\%$. Probably, the high pressure treatment would result in slight packing compactness of the AlF_6 octahedra, which favored the communication between the n electrons of F atoms and suppressed nonradiative deactivation.

Figure R7. photographs of AlF_3 afterglow and measured phosphorescence quantum yield ($\lambda_{\text{ex}} = 285 \text{ nm}$) of original sample and 10 MPa tableting AlF_3 , respectively.

Last, the BX₆ octahedra structure is important for the above CTE. On one hand, the distances between adjacent F atoms were brought close proximity for potential F...F interactions. On the other hand, other species containing abundant F atoms but no such octahedra structure (e.g., NaF or KF), do not show any luminescence.

On the basis of the above analysis, the discussion about STE of α -AlF₃ and its excitation-dependent color-tunable phosphorescence was totally re-written. Please see our revised MS.

Comment 3. Do the author exclude completely a contribution to the long-lived luminescence given by thermally activated delayed fluorescence? The increase of emission intensity with lowering of the temperature makes believe that TADF is not existing in this system, but still it cannot be ruled out just based on T-dependent measurements of the PL and theoretical calculations. The effective existence of the triplet as the emissive species might be seen by other time-resolved techniques like transient absorption/pump-probe spectroscopy or even EPR. Can the author present an argument against the need to carry out also similar more detailed investigations to really explain the observed phenomena?

Reply: Thanks for your expert comments!

First, attempts on transient absorption/pump-probe spectroscopy were carried out, but were failed, probably because of the insoluble and strongly scattering nature of the α -AlF₃ sample.

To further rule out the possibility of TADF, the following evidences were further provided besides T-dependent intensity measurements:

1. The lifetime profile of TADF should contain both short (prompt fluorescence, typically ns) and longer (DF from reversed intersystem crossing) components (Figure R11). However, no such features were observed in our lifetime profiles (no short component).
2. Upon lowering the temperature, the lifetime profile of TADF would be changed greatly (due to the change of delayed fluorescence contribution). However, in our work, the lifetime profiles at 77 K and 298 K were generally similar (only different decay rates, Figure R11).

Figure R11. Phosphorescence lifetime of AlF_3 at room temperature and 77 K condition. (excited by 280, 350 and 390 nm, respectively).

Therefore, we think the luminescence of $\alpha\text{-AlF}_3$ should be phosphorescence, rather than TADF.

Comment 4. The emission in metal halide perovskite is not always due to the STE. In 2D perovskite both excitonic emission and STE can be observed. In 0D it is most of the time purely excitonic. Why do the authors claim that STE is the typical emissive feature in MHPs?

Reply: Thanks very much for your kind reminding!

The related description was revised as follows:

“For the intrinsic luminescence of MHPs, although defect (vacancy)-induced emission has been reported, self-trapped exciton (STE) emission is also well-accepted.”

Comment 5. I do not understand the chemiluminescence measurements. What do the authors want to show with these data?

Reply: Thanks!

The operation details on collection of the luminescence signal with chemiluminescence measurements were given in Figure R12. For such measurements, we could further confirm the long-lived phosphorescence (no wavelength information, only total luminescence intensity) through exactly excitation ceasing.

Figure R12. The instrumental setup for collection of the afterglow images with a chemiluminescence imaging system.

Comment 6. It is not clear why the formation of the triplet should be favored in these systems. There is only one very sibylline sentence that should explain it, but it is indeed, sibylline to me (The excited electron could be stabilized at T_1 state due to orbital parity rule, thus exhibiting effective long-lived phosphorescence, which could be ascribed to the p-p parity forbidden transition). The authors should better clarify why the formation of the triplet is so favored in these systems. In inorganics it is normally the heavy atom effect that drives its formation. Here anyway we are dealing with light elements. The authors may understand that it is not so logical based on generally assumed scientific knowledge to guess that a triplet should form in similar materials. Can the author bring other examples from other inorganic materials forming the triplet and being constituted of light elements? In MHP the PL lifetimes are typically those of singlet states, for example.

Reply: Thanks very much for your expert comment!

In fact, our initial opinion on the luminescence from α -AlF₃ was basically similar to yours (totally light elements, largely different from the existing inorganic phosphors). However, we do acknowledge that our understanding and explanation for the related phenomena were unsatisfied. Thanks to the comments from you and the first reviewer, the related discussions were totally re-written (e.g., see our reply to your comment 2).

In our system, generation of the triplet excitons were determined by following two factors:

1. Heteroatoms (F) are abundant in AlF₃, resulting in through space conjugation of the n electrons of F atoms. Such weak conjugation eventually lead to phosphorescence from AlF₃ (similar to previous nonconventional luminophores, e.g., Zhou Q. et al. Clustering-Triggered Emission of Nonconjugated Polyacrylonitrile. *Small*, **2016**, 12, 6586-6592; Zhou Q., et al. A clustering-triggered emission strategy for tunable multicolor persistent phosphorescence. *Chem. Sci.*, **2020**, 11, 2926-293; Zhang, H.; Tang, B. Z. Through-Space Interactions in Clusteroluminescence, *JACS Au* **2021**, 1, 1805-1814).
2. The rigidification of the AlF₆ octahedra follows the order of AlF₃ (3D network) > KAlF₄ (2D, layered corner sharing) > (NH₄)₃AlF₆ (0D, isolated). In fact, previous references on nonconventional luminophores also exhibited similar stabilizing trends, for example, polyacrylonitrile (Zhou Q., et al. Clustering-Triggered Emission of Nonconjugated Polyacrylonitrile. *Small*, **2016**, 12, 6586-6592), xylitol (Wang Y. Z., et al. Emission and Emissive Mechanism of Nonaromatic Oxygen Clusters. *Macromol. Rapid Commun.*, **2018**, 39, 1800528) and cyanoacetic acid (Fang M., et al. Unexpected room-temperature phosphorescence from a non-aromatic, low molecular weight, pure organic molecule through the intermolecular hydrogen bond. *Mater. Chem. Front.*, **2018**, 2, 2124-2129.). In these works, various intra- or intermolecular interactions resulted in sufficiently rigidified conformations and generate room-temperature phosphorescence.

Comment 7. The authors claim that in organics there are certain moieties that are typically luminescent. I think this is a very arguable statement. It is true that many organic semiconductors featuring such moieties are highly luminescent, but it is also the tendency to

aggregate that drives the luminescence there (the more they can self-aggregate, the more the PL can be quenched. therefore the type of substituents play a major role there). In the end similar affirmations are very relevant to introduce the uniqueness of the fundamental question that they want to raise and to which they want to give an unambiguous answer.

Reply: Thanks very much for your expert comment!

We agree with the reviewer that the initial claim on organics was somewhat absolute. The related part was revised as follows:

“Generally, it is widely accepted that luminescence from organic materials can be ascribed to their core structure in most cases (Fig. 1, together with the substituents), for example, fluorescent materials from xanthene and phosphorescent materials based on carbazole.”

REVIEWER COMMENTS

Reviewer #1 (Remarks to the Author):

First of all, I would like to highlight the enormous effort the authors have put in order to address the comments of both reviewers. They have given a new explanation to their observations, backed up with experiments, that has substantially strengthened the quality and rigour of the work. I agree that the report of a light element based inorganic material showing efficient phosphorescence is a nice addition to the field, and the (updated) mechanism proposed is compatible with the experimental results. Therefore, the work would be suitable for its acceptance in Nature Communications after some further clarification on a couple of points as follows:

- Response to Comment #3 (paragraph before Figure R6): The notion of different sized clusters forming in a nominally 3-dimensional continuous structure should be further discussed. The papers mentioned are on the polymer/molecule level; how does it extend to the systems herein discussed?
- Similarly, in response to comment #8, the authors mention, "In these works, various intra- or intermolecular interactions resulted in sufficiently rigidified conformations and generate room-temperature phosphorescence". Why different sized clusters form in a 3D connected framework?
- Response to Comment #3 (paragraph before Figure R7): What is the Young's modulus of the material? It is probably in the order of several to tens of GPa. I do not think that 10 MPa will do much in the way of strain on the atomic level. The observation is more likely due to changes in morphology and/or light outcoupling.
- The response to my Comment #10 is still incorrect.

Reviewer #2 (Remarks to the Author):

The manuscript has been substantially revised, which has improved both the science reported within and the quality of the presentation. Still, I find some details that raise concerns and confusion to the reader.

Here a list:

- Figure 2: the caption says "Luminescence properties of α -AlF₃ (calcinated from AlF₃·3H₂O)", but in the text the calcination is reported as a second proof of the purity of the sample, to exclude that the emissive properties detected on a previously mentioned sample (not clear how this last one was prepared) effectively come from the octahedral inorganic scaffold and not from impurities. Please try to be very specific in the captions on the source of your images. Given the number of different samples and species you are investigating here, it is of the utmost importance to be absolutely clear in every single detail.
- the authors talk about fluoride n-electrons conjugation. Is it really correct to define conjugation similar van der Waals interactions? Conjugation is normally used for pi electrons clouds and has also a significant effect on absorption properties (the more conjugated, the more red shifted absorption)
- although the logic behind the choice of the different inorganic scaffolds has been largely clarified in reply to Reviewer 1 concerns, it is still difficult to catch the sense of going from Al, chosen as the lightest central element in BX₆ octahedra, up to gallium, cadmium and indium. please provide a consequential justification, also considering the increased atomic weight.
- In the discussion part, I would further try to add a summarizing scheme that illustrates the overall logic of the investigations carried out.

Manuscript ID: NCOMMS-21-37768A-Z

Title: Multicolor Ultralong Phosphorescence from Perovskite-Like Octahedral α -AlF₃

Reviewer #1

Remarks to the author:

First of all, I would like to highlight the enormous effort the authors have put in order to address the comments of both reviewers. They have given a new explanation to their observations, backed up with experiments, that has substantially strengthened the quality and rigour of the work. I agree that the report of a light element based inorganic material showing efficient phosphorescence is a nice addition to the field, and the (updated) mechanism proposed is compatible with the experimental results. Therefore, the work would be suitable for its acceptance in Nature Communications after some further clarification on a couple of points as follows:

Reply: Many thanks for the reviewer's professional comments and appreciation on our work.

Comment 1. Response to Comment #3 (paragraph before Figure R6): The notion of different sized clusters forming in a nominally 3-dimensional continuous structure should be further discussed. The papers mentioned are on the polymer/molecule level; how does it extend to the systems herein discussed?

Reply: Thanks very much for your valuable comments!

To further investigate the influence of clustering effect from AlF₆ octahedra on luminescence, larger sized clusters were calculated with quantum mechanics and molecular mechanics (QM/MM) method. As can be seen from Figure R1 below, the energy gap decreased as the number of AlF₆ octahedra increasing (increasing the sizes of the clusters). Therefore, different clustering states may result in varied energy gaps, thereby excitation-dependent and color-tunable phosphorescence.

Figure R1. Schematic illustration of the AlF₆ clusters (isolated, layered and cubic corner sharing AlF₆ octahedra) and corresponding calculated energy gap.

The above information was added in the revised MS as follows (Lines 15-19 in Page 14):

“Moreover, as further calculated with the quantum mechanics and molecular mechanics (QM/MM) method, the energy gap decreased as increasing the number of AlF₆ octahedra (increasing the sizes of the clusters, Supplementary Fig. 30). Therefore, different clustering states may result in varied energy gaps (Fig. 5d), thereby excitation-dependent and color-tunable phosphorescence.”

Comment 2. Similarly, in response to comment #8, the authors mention, “In these works, various intra- or intermolecular interactions resulted in sufficiently rigidified conformations and generate room-temperature phosphorescence”. Why different sized clusters form in a 3D connected framework?

Reply: Thanks very much for your expert comments!

During our last reversion, the multicolor emission from α -AlF₃ was ascribed to clusterization-triggered emission, i.e., from varied emission centers (Fig. 5d). Here, the definition of ‘cluster’ did not correspond with the real size of the sample, but the multi emission centers generated from effective through-space interaction of n electrons in the clustering state.

To avoid potential misleading, Figure 5 was revised as follows:

Figure R2. Revised Figure 5.

Comment 3. Response to Comment #3 (paragraph before Figure R7): What is the Young's modulus of the material? It is probably in the order of several to tens of GPa. I do not think that 10 MPa will do much in the way of strain on the atomic level. The observation is more likely due to changes in morphology and/or light outcoupling.

Reply: Thanks very much for your expert comment!

First, the exact Young's modulus of AlF₃ (melting point: 1290 °C) was not accessed from the existing text books or references, but it was estimated as tens to hundreds of GPa, since the Young's modulus is positively correlated with the melting temperature (Al₂O₃: melting point, 2072 °C, 366 GPa; NaCl: melting point, 800.7 °C, 34 GPa).

Second, Young's modulus represents the interatom bonding force, which is notably larger than the intermolecular one (e.g., Van-der Waals interaction or hydrogen bonding). The applied pressure (10 MPa, in fact, should be 900 MPa due to previous incorrect calculation) was still largely smaller than the Young's modulus. Therefore, the high pressure applied to the sample here was not enough to impact strain at the atomic level. The previous description in our MS (the high pressure treatment would result in slight packing compactness of the AlF_6 octahedra) was indeed misleading.

Third, the weak interaction between AlF_6 octahedra in $\alpha\text{-AlF}_3$ was further investigated with Atoms in Molecules (AIM) analysis. As shown in Figure R3 below, in addition to traditional bond path in the octahedron (Al-F), there is also plenty of through-space interaction path ($\text{F}\cdots\text{F}$ interaction) as the AlF_6 octahedra system extended. High pressure treatment is expected to strengthen the weak intermolecular Van-der Waals interaction, thereby slightly higher emission brightness of $\alpha\text{-AlF}_3$.

Figure R3. AIM analysis of layered and cubic corner sharing AlF_6 octahedra. The bond (BCP), ring (RCP) and cage (CCP) critical points represent the extreme points of electron density on the bond paths, centers of rings, and enclosed space formed by rings, respectively. Through space interaction (TSI) paths were indicated by orange dash lines.

To further illustrate such issue, the related contents were revised as follows:

“The weak interaction between AlF_6 octahedra in $\alpha\text{-AlF}_3$ was further investigated with Atoms in Molecules (AIM) analysis. As shown in Supplementary Fig. 31, in addition to traditional

bond path in the octahedron (Al-F), there is also plenty of through-space interaction path (F...F interaction) as the AlF_6 octahedra system extended. Experimentally, upon high pressure treatment (900 MPa) to strengthen the weak intermolecular Van-der Waals interaction, the emission brightness of $\alpha\text{-AlF}_3$ was increased somewhat (Fig. 5e, Φ_{P} increase from $\sim 4.22\%$ to $\sim 5.46\%$), while the multicolor emission profiles of $\alpha\text{-AlF}_3$ were not disturbed (Supplementary Fig. 19). Therefore, such interactions are expected to facilitate electron communications between the n electrons of F atoms and rigidify the system for efficient phosphorescence.”

Besides, for better illustrating the high pressure treatment, Fig. 5e and 5f were re-designed as follows:

Figure R4. Photograph of AlF_3 afterglow with a flower pattern, consisting of normal $\alpha\text{-AlF}_3$ and their pressurized sample, respectively.

Comment 4. The response to my Comment #10 is still incorrect.

Reply: Thanks very much for your kind reminding!

The related description was revised again as follows (Lines 5-7 in Page 10):

“The intrinsic emission of MHPs could be originated from free, bound and self-trapped excitons. Among them, self-trapped exciton (STE) emission is a well-accepted mechanism to account for the broadband and large-stokes shift emission.”

Reviewer 2

Remarks to the author: The manuscript has been substantially revised, which has improved both the science reported within and the quality of the presentation. Still, I find some details that raise concerns and confusion to the reader.

Reply: Thanks very much for your professional suggestions on this work.

Comment 1. Figure 2: the caption says "Luminescence properties of α -AlF₃ (calcinated from AlF₃·3H₂O)", but in the text the calcination is reported as a second proof of the purity of the sample, to exclude that the emissive properties detected on a previously mentioned sample (not clear how this last one was prepared) effectively come from the octahedral inorganic scaffold and not from impurities. Please try to be very specific in the captions on the source of your images. Given the number of different samples and species you are investigating here, it is of the utmost importance to be absolutely clear in every single detail.

Reply: Thanks very much for your kind reminding!

We agree with the reviewer that the contents here were frustrated, which may be originated from several rounds of revisions.

In Figure 2, the luminescence properties of α -AlF₃ were presented, since it is the direct example of AlF₆ octahedra (corner-shared). Also, it is the most stable phase of AlF₃. But in our initial investigations, we didn't have the α -AlF₃ sample at hand. We thus prepared α -AlF₃ through calcination of β -AlF₃·3H₂O (Fig. S4). Of course, the calcination process was also capable of removing possible organic impurities in β -AlF₃·3H₂O.

To avoid potential misleading, the contents about the proof of the purity were revised as follows:

"To exclude the potential influence from trace impurities, direct synthesis of AlF₃ through exposing aluminum metal of the highest purity available to HF vapor was carried out. As expected, similar emission properties were also obtained (Supplementary Figs. 6-7), confirming that the luminescence was exactly from AlF₃. Furthermore, the purchased and as-prepared samples were processed by calcination, ball milling, and acid-washing, and no appreciable change of the luminescence property was received (Supplementary Figs. 8-10)."

Comment 2. the authors talk about fluoride n-electrons conjugation. Is it really correct to define conjugation similar van der Waals interactions? Conjugation is normally used for pi electrons clouds and has also a significant effect on absorption properties (the more conjugated, the more red shifted absorption).

Reply: Thanks very much for your expert comment.

We agree with the reviewer that “conjugation” here may be inappropriate. To be more accurate, we changed “conjugation” to interaction.

Besides, the related part was revised as follows (Lines 5-7 in Page 14):

“Therefore, there is possible van der Waals F...F interaction, thus leading to effective electron cloud overlap of the n electrons for $n \rightarrow \sigma^*$ transition.”

Comment 3. although the logic behind the choice of the different inorganic scaffolds has been largely clarified in reply to Reviewer 1 concerns, it is still difficult to catch the sense of going from Al, chosen as the lightest central element in BX_6 octahedra, up to gallium, cadmium and indium. please provide a consequential justification, also considering the increased atomic weight.

Reply: Thanks for your expert comments!

Here, we wanted to investigate whether the BX_6 octahedra can be a potential core structure for the luminescent inorganic materials. Therefore, we started with $\alpha\text{-AlF}_3$ (3D) featuring the smallest metal center (Al) and ligand (F) and 3D corner shared BX_6 octahedra. Then, the dimension, ligand, and the metal center of the BX_6 octahedra were altered for investigation.

In light of your kind suggestion, Figure 2 was further revised for clear illustration of the sense as follows:

Figure R5. Revised Figure 2.

Comment 4. In the discussion part, I would further try to add a summarizing scheme that illustrates the overall logic of the investigations carried out.

Reply: Thanks for your valuable comments!

We added the following chart in Page 17 of the revised MS to illustrate the overall logic of this work.

Figure R6. Summary on the BX₆ octahedra-based luminescence.

REVIEWERS' COMMENTS

Reviewer #2 (Remarks to the Author):

I think the manuscript is now ready for publication. All comments from the 2 reviewers were adequately addressed.

It is a very complete and detailed work, with a lot of precious information and new insights.

Manuscript ID: NCOMMS-21-37768B

Title: Multicolor Ultralong Phosphorescence from Perovskite-Like Octahedral α -AlF₃

Reviewer #2

Remarks to the author:

I think the manuscript is now ready for publication. All comments from the 2 reviewers were adequately addressed.

It is a very complete and detailed work, with a lot of precious information and new insights.

Reply: Thanks very much for your positive comment!